



# CO$_2$/$^3$He ratios reveal the presence of mantle gas in the CO$_2$-rich groundwaters of the Ardenne massif (Spa, Belgium)

Agathe Defourny[1,2], Pierre-Henri Blard[3,4], Laurent Zimmermann[3], Patrick Jobé[2], Arnaud Collignon[2], Frédéric Nguyen[1], and Alain Dassargues[1]

[1]Urban and Environmental Engineering, University of Liège, Belgium
[2]Water Resource Departement, Spadel S.A., Belgium
[3]CRPG, Université de Lorraine, CNRS, UMR7358, Nancy, France
[4]Laboratoire de Glaciologie, ULB, Brussels, Belgium

**Correspondence:** Agathe Defourny (adefourny@uliege.be)

**Abstract.** Natural CO$_2$-rich groundwaters of eastern Belgium have been known for centuries although the exact origin of the gas they contained was still unclear. This paper presents the results of a sampling campaign in the area (Spa, Stoumont, Malmedy) where 30 samples of both carbogazeous and non-carbogazeous groundwaters have been analyzed for major elements, CO$_2$ content and carbon isotopic composition. Among them, 13 samples were analyzed for $^3He/^4He$ and $^4He/^{20}Ne$ ratios.

The combination of $\delta^{13}C$ and $^3He$/CO$_2$ ratios have shown with a high level of confidence that the CO$_2$ in groundwater from springs and boreholes has a mantellic origin, and can very likely be attributed to the degassing of the neighboring and still buoyant Eifel mantle plume, located at a distance of 100 km eastwards. The identity and nature of the deep-rooted fractures that act as CO$_2$ transport pathway to the surface are still to be clarified, but several major thrust faults exist in the Rhenish Massif and could connect the Eifel volcanic field with the studied area.

## 1  Introduction

CO$_2$-rich groundwaters have always been a very fascinating geomanifestation and their presence in a region is often the trigger of a strong economic and touristic activity. Lately, a better assessment of CO$_2$ circulation modes in the sub-surface has gained in interest, as it is important to finely document their contribution to the natural budget of atmospheric CO$_2$. Understanding the saturation state of CO$_2$ in groundwaters and in geological reservoirs is also important for CO$_2$ storage projects. Moreover,

in areas where CO$_2$-rich groundwater is bottled as mineral water, it is essential to have a complete understanding of the whole system, to ensure sustainable exploitation of the resource. The presence of CO$_2$ in groundwater - in excess compared to the atmospheric equilibrium - can result from different phenomena, the most common ones being a direct contribution from the mantle, the dissolution of carbonate rocks, or an organic origin  (Agnew 2018). The discharge of CO$_2$ from deep geological structures to the surface is always the result of a specific geological context which involves a source of CO$_2$ at depth and an

intricate system of faults acting as transport pathways to the local groundwaters, considered as the final receptor.





This study focuses on the dozen of $CO_2$-rich groundwater springs that exist in the Ardenne Massif, in Eastern Belgium. The most famous ones are located in the small city of Spa, whose springs have been known since the Roman Empire. The name of the Spa town has then become famous thanks to the developement of thermalism in the 19th century and is now used worldwide

to refer to wellness and bathing activities. These groundwaters yield natural springs at the surface, but they are also exploited as mineral water from boreholes. Their $CO_2$ content ($2 \pm 0.5$ g/l) makes them slightly acidic (pH around 5.7). They have a high content in iron ($17 \pm 10$ mg/l on average, but up to 50 mg/l). They however bear a relatively low TDS (Total Dissolved Solid) with a dry residue ranging between 80 mg/l and 160 mg/l in comparison with other naturally sparkling mineral waters bottled in other European geological contexts (e.g., 940 mg/l for San Pellegrino, 1100 mg/l for Badoit, 3325 mg/l for Vichy Celestins).


Although these springs have been bottled for centuries and studied for many years, the origin of their high $CO_2$ content has not been established already. Helium isotopes, elemental $^4He/^{20}Ne$ and $CO_2$ isotopic composition of dissolved gas is a powerful tool to identify the sources of these gas ( (Sano and Marty 1995; Karolyte et al. 2019; Gilfillan et al. 2019). In this paper, we present the $^3He$, $^4He$, $^{20}Ne$ and $CO_2$ concentrations measured in 13 groundwater samples of the Spa and Bru areas (Ardennes,

Belgium), together with hydrochemical analysis on major elements for 30 samples, to identify the origin of dissolved $CO_2$ in groundwater and to explore the potential hydro-connection with the Eifel Volcanic Fields (Western Germany), where similar $CO_2$ rich groundwaters are found.

## 2   Geological context

### 2.1   Regional geology

$CO_2$-rich mineral waters from eastern Belgium are located in the Rhenish Massif, which is part of the Rhenohercynian fold belt  (Vanbrabant, Braun, and Jongmans 2002). This massif extends through western Germany, eastern Belgium, Luxembourg, and a part of France, as shown in Figure 1. The Rhenish Massif is dominated by Paleozoic rocks and is separated into two parts by the Rhine Graben. The Western part, the Ardennes (eastern Belgium), is bordered in the north by the Midi-Eifel thrust fault. In the Ardenne region, the Rhenish Massif is dominated by the Ardenne Allochtone, consisting of a Cambro-Ordovician

basement unconformably overlain by Devono-Carboniferous sandstones and limestones  (Barros et al. 2021) (Figure 1).

The present regional geology is the result of several stages. The oldest rocks that can be observed in Belgium are found in the Ardennes region, they consist in Cambrian to Ordovician sediments deposited in deep platform marine environments. They mainly consist of fine clays in alternation with sandstones. During the Late Silurian, the Caledonian orogeny faulted and

folded these layers and induced a strong metamorphism. Compression and shearing of claystone produced a well-expressed schistosity, while sandstone evolves into quartzites. The metasediments observed today are therefore an alternation of clays and sandstones metamorphosed into slates ('phyllades') and quartzites, which are called 'quartzophyllades' in the region. Then, after an emersion and erosion period, sedimentary deposition started again during the Lower Devonian, in unconformity over the folded rocks. During the Lower Devonian period, limestones were deposited in vast carbonate platforms. Then, sandstones





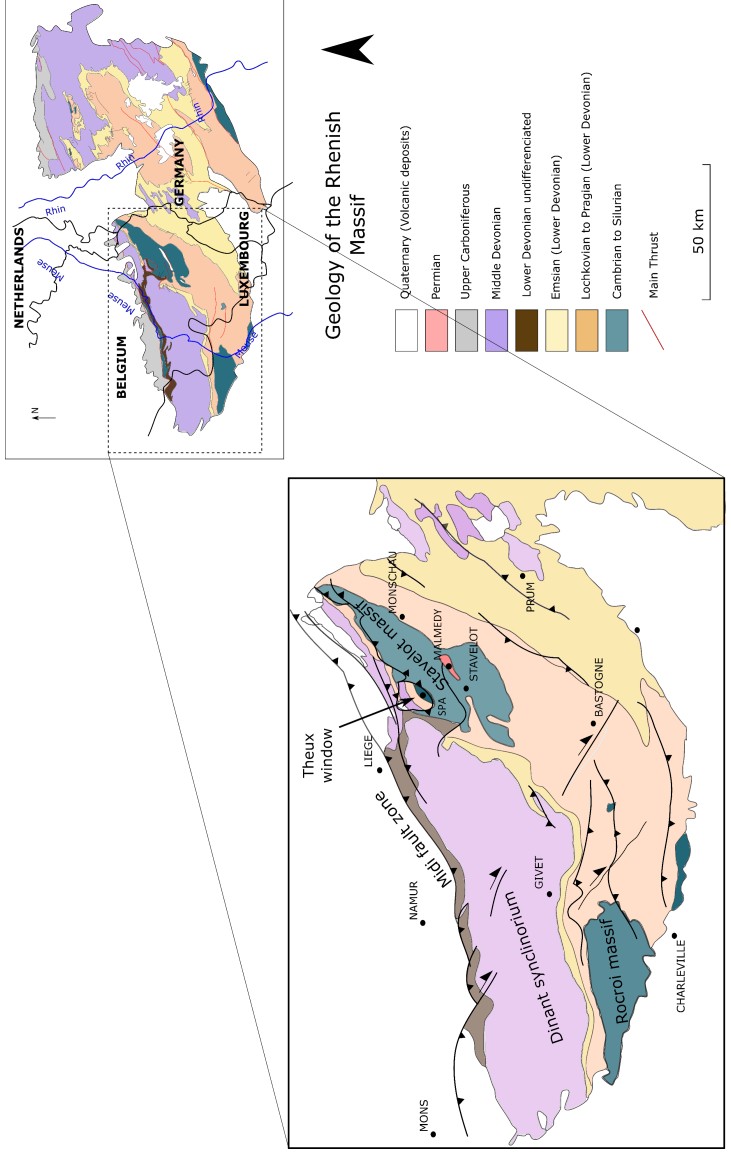

**Figure 1.** Simplified geology of the Ardennes Allochton and its localisation in western Europe and in the Rhenish Massif, modified after Fielitz and Mansy 1999.

accumulated in coastal detrital environments during the Devonian.

At the end of the Devonian, the Variscan orogeny took place, inducing another phase of tectonic deformation. During this orogeny, the Synclinorium of Dinant and the Ardenne Massif were displaced several kilometers northward. This (latter) unit constitutes the Ardennes Allochthone, a great anticlinal ensemble that is limited to the north by the Midi-Eifel thrust fault.

Three Cambrian massifs are identified within the Ardenne Allochtone: the Rocroi, the Givonne and the Stavelot massifs. The





studied springs are located within the Stavelot Massif and at its border with the Dinant Synclinorium. The Graben of Malmedy, filled with Permian conglomerates, developed in the center of the Stavelot Massif with an SW-NE orientation is separating the Stavelot Massif into two parts.

## 2.2  CO$_2$-rich groundwater springs in the Rhenish Massif and the Eifel volcanif fields

Numerous occurrences of CO$_2$-rich groundwater springs are recorded in the Rhenish Massif, as shown in Figure 2. CO$_2$-rich groundwaters present in Belgium are cold waters (12 °C on average). They are fed by the recharge from local or regional precipitations, as confirmed by $^{18}$O and $^{2}$H isotopic measurements  (Barros et al. 2021). The system is dominated by meta­morphised sedimentary rocks and the aquifer zones lie in the first hundreds of meters of these deposits, in the fractured and

weathered parts.

Naturally sparkling groundwaters are bottled by three different companies in the area (Bru-Chevron and Spa Monopole in Belgium, and Gerolsteiner Brunnen in Germany. While it has been proven for a long time that the dissolved CO$_2$ present in the springs of western Germany was the result of mantle degassing, this has not been confirmed for the Belgian springs yet  (May,

Hoernes, and Neugebauer 1996; Aeschbach-Hertig et al. 1996; Barros et al. 2021).

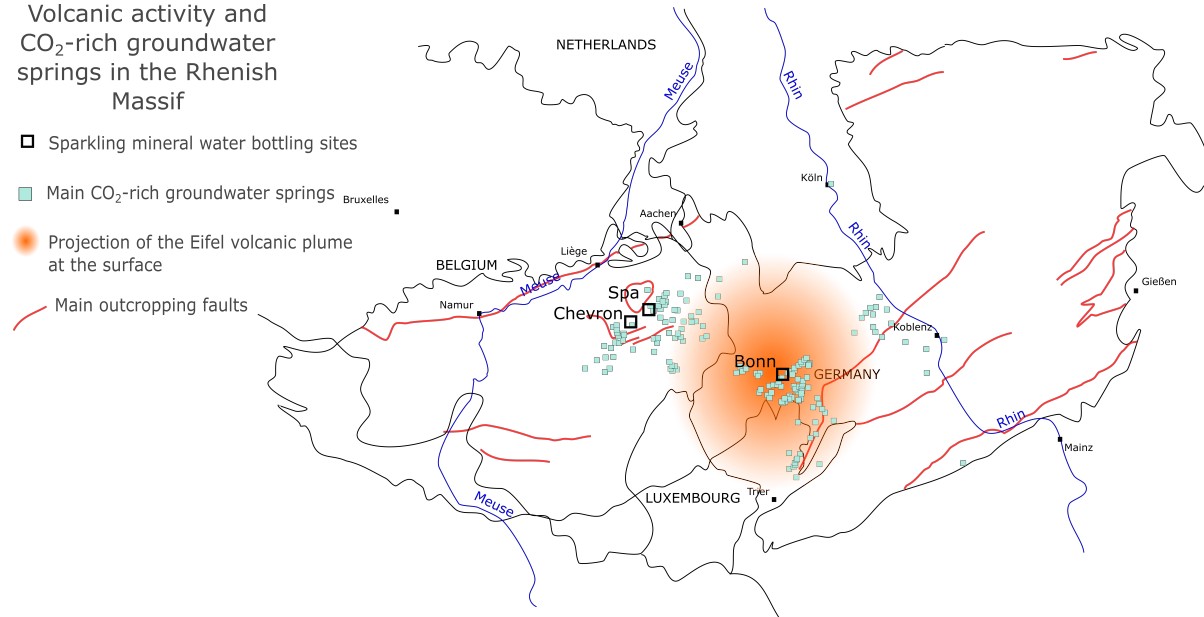

**Figure 2.** Occurrences of CO$_2$-rich groundwater springs in the Rhenish Massif, together with the main bottling sites. The springs locations were compiled from  May, Hoernes, and Neugebauer 1996; Bräur et al. 2013 and intern data from Spadel. The projection of the Eifel volcanic plume is depicted after  Bräur et al. 2013.





However, the proximity between the two gas-rich groundwater areas, their common geological context, and the scale of the Eifel volcanism suggests that the gas they contain might have a common origin. In the Eifel area, volcanic activity has occurred during the Quaternary. It is thought that the lower mantle upwelling under central Europe may feed smaller upper-

mantle plumes (Goes, Spakman, and Bijwaard 1999). Indeed, small plume structures have been identified below the Eifel area by teleseismic tomography (see Ritter et al. 2001) and a recent study by Kreemer et al. (2020) has shown that this plume was still buoyant (Kreemer, Blewitt, and Davis 2020).

## 3   Sampling and analysis

For this study, we collected water samples from natural springs and wells at 30 different locations distributed within an area of (30 km x 20 km), belonging either to Spa, Bru or Malmedy regions (Table 1, Figure 3). Each of them has been analyzed for major and traces elements, physicochemical parameters, dissolved gases ($O_2$, $CO_2$, He, Ne) and carbon isotopes ($\delta^{13}C$). During a second campaign, we resampled 13 of these 30 sites, with the specific purpose of analyzing their $^3He/^4He$ and $^4He/^{20}Ne$ isotopic ratios. Samples were stored into copper tubes clamped on both sides to prevent any degassing or air contamination.


Major and traces elements, together with physicochemical parameters and dissolved $O_2$ and $CO_2$, were measured at the Spadel hydrochemistry laboratory, following the specific certified procedures: ISO 10523 (pH), ISO 7888 (electrical conductivity), ISO 10304-1 (chlorides, sulfates, and nitrates), ISO 9963 (bicarbonates), ISO 17289 (dissolved oxygen) and ISO 17294 (ICP-MS for the rest of the elements). Dissolved $CO_2$ was quantified by mineral sequestration via a barium sulfate saturated

solution, followed by an inverse titration. Carbon isotopic ratios were measured by a private lab using isotope-ratio mass spectrometry (IRMS) and gas-chromatography isotope-ratio mass-spectrometry (GC-IRMS). Finally, the dissolved concentrations of helium isotopes were analyzed at the CRPG of Nancy (CNRS-UMR 7358) by static vacuum mass spectrometry after vacuum extraction and purification according to Zimmermann and Bekaert (2020) and Zimmermann et al. (Zimmermann and Bekaert 2020; Zimmermann, Furi, and Burnard 2015). $^4He/^{20}Ne$ ratio was also measured in the same water aliquots, with a

quadrupole installed on the extraction line. Instrument sensitivity was determined against a gas standard having an atmospheric composition. Tap water that had been placed in equilibrium with the atmosphere was also analyzed for comparison (Table 1).

Figure 3 shows the location of each sample in the local geology.

## 4   Results

Table 1 presents the gas results measured in our 30 new samples. Data from Victoriaquelle (VQ) and Schwefelquelle (SQ) are from (Marty et al. 2020). Data from the Laacher See Mofetta (LaS) and Wehr 10 well (W10) (Bräur et al. 2013) are also included (i.e., a mofetta is a fumarole discharging mostly carbon dioxide). All these values can be compared to the Mid-ocean





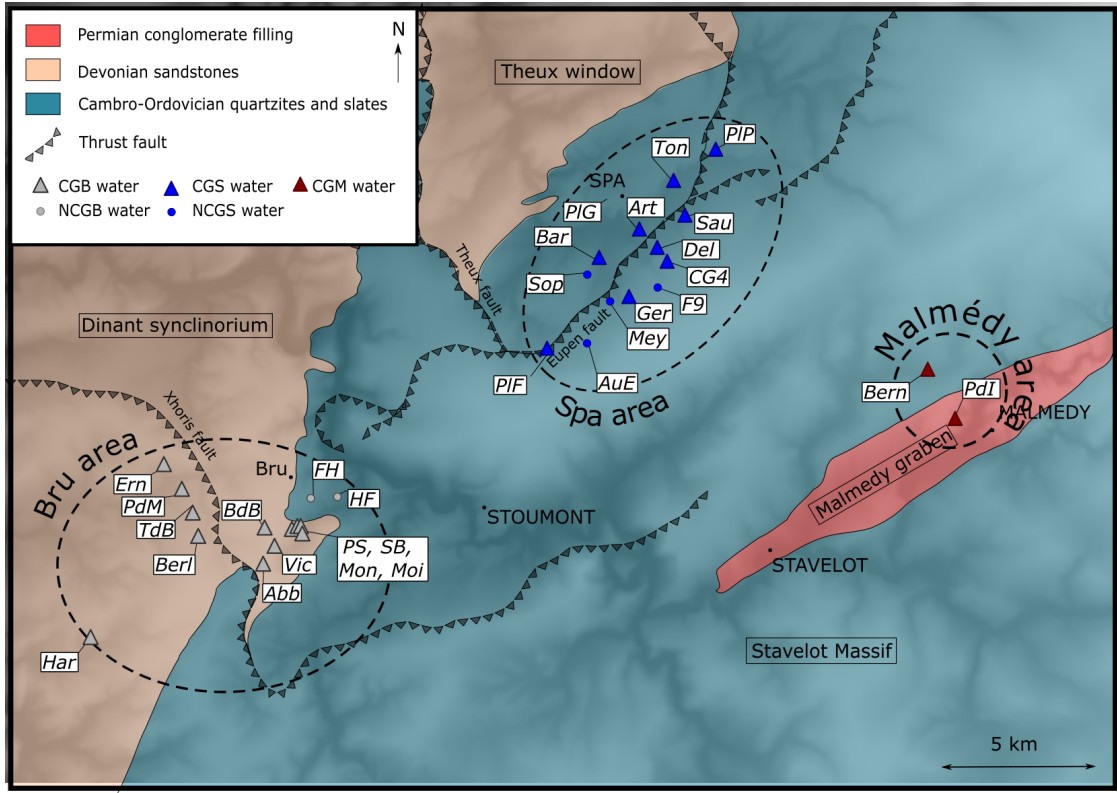

**Figure 3.** Location of each sample in regard with the main geological features.

ridge basalts (MORB) endmember, which is commonly accepted to represent the signature of the upper mantle and have been studied in detail by Graham (2002) (Graham 2002).






| | Type | | pH | CE | Ca | Mg | Na | K | Cl | SiO$_2$ | HCO$_3$ | SO$_4$ | Fe | Mn | CO$_2$ | O$_2$ | $\delta^{13}C$ DIC ±0.3‰ | R/Ra | ± | $^4$He/$^{20}$Ne | ± |
|---|---|---|---|---|---|---|---|---|---|---|---|---|---|---|---|---|---|---|---|---|---|
| | | | | µS/cm | mg/l | mg/l | mg/l | mg/l | mg/l | mg/l | mg/l | mg/l | mg/l | mg/l | g/l | mg/l | ‰VPDB | [] | [] | [] | [] |
| BdB | CGB | Well | 5.2 | 301 | 21 | 15.63 | 6.92 | 1.67 | 3.3 | 20.84 | 143 | <5 | 16.19 | 1.39 | 2.68 | 2.47 | -2.4 | 2.7 | 0.02 | 150.63 | 9.55 |
| Moi | CGB | Spring | 5.4 | 424 | 25.7 | 24.93 | 8.51 | 1.71 | 4.2 | 17.58 | 234 | <5 | 25.15 | 1.86 | 1.81 | 2.89 | -2.9 | | | | |
| Mon | CGB | Well | 5.6 | 556 | 68.4 | 23.24 | 13.1 | 1.75 | 5 | 21.24 | 315 | <5 | 7.05 | 1.3 | 2.02 | 2.86 | -3.4 | 2.42 | 0.02 | 154.35 | 5.4 |
| SB | CGB | Spring | 5.1 | 232 | 12.3 | 11.43 | 7.46 | 0.97 | 3.9 | 20.98 | 91 | <5 | 13.93 | 1.14 | 2.15 | 3.58 | -3.4 | | | | |
| Abb | CGB | Well | 5.9 | 274 | 23.4 | 12.91 | 8.7 | 0.78 | 4 | 30.9 | 152 | 7.5 | 6.77 | 1.36 | 0.7 | 9.49 | -8.8 | 2.61 | 0.02 | 68.16 | 2.7 |
| PdM | CGB | Well | 5.9 | 223 | 23.9 | 10.76 | 5.65 | 1.07 | 12.8 | 7.07 | 100 | <5 | 7.46 | 0.48 | 0.77 | 6.95 | -8.7 | 2.42 | 0.01 | 33.19 | 1.54 |
| PS | CGB | Well | 5.3 | 454 | 23.7 | 24.1 | 4.67 | 1.5 | 3.4 | 17.7 | 231 | <5 | 35 | 2.11 | 3.45 | 0.33 | -2.4 | 2.5 | 0.01 | 348.68 | 16.6 |
| Art | CGS | Well | 5.3 | 100 | 4.3 | 3.39 | 4.01 | 0.42 | 5.4 | 7.2 | 23 | 16.7 | 5.46 | 0.13 | 0.91 | 5.58 | -7.6 | 1.15 | 0.01 | 69.64 | 2.12 |
| CG4 | CGS | Well | 5.8 | 212 | 16.7 | 9.29 | 8.35 | 1.46 | 4.5 | 7.5 | 113 | <5 | 7.14 | 0.24 | 1.22 | 6.71 | -5.7 | | | | |
| PIP | CGS | Well | 5.4 | 257 | 15.9 | 9.32 | 4.97 | 0.84 | 3.1 | 10.04 | 137 | <5 | 22.06 | 0.38 | 2.35 | 0.76 | -3.8 | | | | |
| PIF | CGS | Well | 5.8 | 312 | 30.4 | 10.16 | 20.46 | 6.48 | 3.6 | 11.25 | 169 | 11.8 | 4.04 | 0.16 | 1.27 | 2.46 | -4.9 | 1.85 | 0.02 | 45.19 | 1.65 |
| HF | NCGB | Well | 6.5 | 168 | 15 | 3.76 | 12.04 | 0.83 | 5.3 | 16.58 | 63 | 17 | 0.26 | 0.53 | 0.14 | 8.11 | -15.7 | 1.87 | 0.01 | 4.05 | 0.16 |
| FH | NCGB | Well | 6.5 | 169 | 21.2 | 2.69 | 7.67 | 1 | 4.1 | 14 | 64 | 16.1 | 0.098 | 0.43 | 0.01 | 9.57 | -15.2 | 1.73 | 0.01 | 1.66 | 0.07 |
| AuE | NCGS | Well | 5.8 | 52 | 3.8 | 1.98 | 2.27 | 0.2 | 3.1 | 7.05 | 12 | 7.8 | 2.07 | 0.05 | 0.09 | 2.52 | -22.6 | 1.65 | 0.01 | 2.66 | 0.12 |
| F9 | NCGS | Well | 5.8 | 38 | 3.3 | 0.78 | 2.46 | 0.26 | 3.2 | 7.13 | 8 | <5 | 0.85 | 0.02 | 0.02 | 4.06 | -27.0 | | | | |
| Sop | NCGS | Well | 6.4 | 92 | 8.4 | 2.05 | 6.26 | 0.49 | 5.2 | 13.57 | 32 | 5.7 | <0.005 | 0.005 | 0.12 | 6.13 | -18.3 | 1.06 | 0.02 | 4.79 | 0.24 |
| Mey | NCGS | Well | 6 | 64 | 4.3 | 1.8 | 2.68 | 0.28 | 3.6 | 6.77 | 19 | 7.8 | 2.29 | 0.05 | | 5.72 | -17.6 | | | | |
| PGd | CGS | Spring | 5.7 | 809 | 42.7 | 35.47 | 71.52 | 4.75 | 58.2 | 54.56 | >204 | 12.8 | 17.35 | 1.81 | 2.64 | 0.56 | -4.5 | | | | |
| Ton | CGS | Spring | 5.1 | 195 | 11.9 | 7.24 | 6.57 | 0.79 | 10.7 | 16.73 | 71 | <5 | 13.33 | 0.92 | 2.24 | 0.84 | -3.9 | | | | |
| Bar | CGS | Spring | 5.3 | 239 | 12.1 | 9.77 | 17.36 | 1.16 | 5.6 | 27.22 | 114 | <5 | 9.95 | 1.2 | 2.10 | 0.77 | -4.2 | 1.1 | 0.02 | 106.93 | 3.12 |
| Ger | CGS | Spring | 5.7 | 185 | 15.3 | 7.55 | 9.74 | 1.23 | 5.9 | 11.61 | 92 | <5 | 4.15 | 0.19 | 1.28 | 3.63 | -5.9 | | | | |
| Sau | CGS | Spring | 5.5 | 336 | 27.9 | 13.38 | 12.26 | 1.33 | 5.5 | 10.04 | 182 | <5 | 31.95 | 0.7 | 2.14 | 2.3 | -3.9 | | | | |
| Del | CGS | Spring | 5.6 | 298 | 22.7 | 14.15 | 16.03 | 2.12 | 3.5 | 10.94 | 170 | <5 | 6.54 | 0.17 | 1.39 | 3.61 | -4.7 | 0.92 | 0.01 | 278.02 | 11.42 |
| TdB | CGB | Spring | 5.9 | 856 | 108.8 | 46.28 | 20.12 | 1.8 | 4.4 | 20.12 | 539 | <5 | 51.91 | 2.76 | 3.12 | 0.27 | -2.1 | | | | |
| Vic | CGB | Spring | 5.2 | 287 | 19.5 | 13.76 | 7.84 | 1.19 | 3.8 | 23.36 | 131 | <5 | 37.52 | 2.72 | 2.08 | 1.26 | -2.9 | | | | |
| Ern | CGB | Spring | 5.8 | 863 | 46.6 | 30.3 | 104.49 | 5.09 | 59.6 | 9.77 | 414 | <5 | 8.34 | 0.76 | 2.32 | 1.82 | -2.9 | | | | |
| Bem | CGM | Spring | 5.4 | 167 | 10.3 | 6.24 | 3.21 | 0.52 | 5.8 | 8.13 | 66 | <5 | 13.14 | 0.47 | 1.3 | 3.92 | -4.0 | | | | |
| PdI | CGM | Spring | 6.3 | 2440 | 418.6 | <0.4 | 68.71 | 5.78 | 37.8 | 12.61 | 1736 | 14.6 | 29.3 | 6.26 | 2.89 | 1.03 | -1.8 | | | | |
| Berl | CGB | Spring | 6.3 | 323 | 28.2 | 8.56 | 7.16 | 1.2 | 14.2 | 8.8 | 165 | 5.2 | 25.49 | 2.1 | 0.95 | 5.49 | -3.4 | | | | |
| Har | CGB | Spring | 5.8 | 672 | 38.4 | 37.2 | 45.02 | 2.88 | 26.6 | 33.3 | 366 | <5 | 14.86 | 0.73 | 2.49 | 2.54 | -2.9 | | | | |
| VQ | CGE | Spring | | | | | | | | | | | | | | | -2.1 | 4.4 | | 1880 | |
| SQ | CGE | Spring | | | | | | | | | | | | | | | -2.0 | 4.5 | | 1520 | |
| LaS | Gaz | Mofette | | | | | | | | | | | | | | | | 5.4 | | 38 | |
| W10 | CGE | Well | | | | | | | | | | | | | | | | 5.6 | | 1380 | |
| MORB | | | | | | | | | | | | | | | | | | 8.1 | | 1000 | |
| Tap | | | | | | | | | | | | | | | | | | 0.971 | 0.052 | 0.292 | 0.013 |

**Table 1.** Analysis results for the 30 samples. CGB = Carbogazeous water from Bru area; CGS = carbogazeous water from Spa area, NCGB= non-carbogazeous water from Bru area, NCGS = non-carbogazeous water from Spa area, CGM = carbogazeous water from Malmedy area. R/Ra is the He isotopic ratio expressed with regards to ration in air. VPDB is the Viennea PeeDee Belemnite standard reporting the abundance of carbon. DIC stands for dissolved inorganic carbon.



All the data obtained from these groundwater samples of Spa, Bru and Malmedy (cations, anions, dissolved $CO_2$, $^3He/^4He$ and $^4He/^{20}Ne$ ratios) are available in Table 1.

## 4.1 Chemical composition of groundwater and dissolved gases

The Piper diagram presented in Figure 4 shows the relative proportion of cations and anions for each sample. The Figure shows
a calcium magnesium bicarbonate type for most samples. Regarding the cations (Ca, Mg, Na+K), the majority of the samples have rather closed compositions and display globally balanced composition, the less abundant cation representing in any case at least 10 pour cent of the total. CGB samples compensate a smaller proportion of Na+K with a bigger proportion of Mg, compared to CBS samples. One exception to this balanced composition in cations, is the *PdI* sample, which is located (in the Piper diagram) on a near-pure calcium pole, probably because it was sampled located in a zone influenced by the carbonate-rich
conglomerate of the Graben of Malmedy (shown in Figure 3). The distinction between CG and NCG groundwater samples is much clearer looking at their anions compositions. There is a significant relative enrichment of CG samples in bicarbonates, due to the presence of dissolved $CO_2$. Only the samples *Art* and *PlG* range out of this carbonate-rich cluster. However this lower carbonate content is only relative as the sample *Art* has a much higher concentration in sulfates (16.7 mg/l whereas most samples are <5 mg/l), presumably from a local geological origin, and the sample *PlG* has a higher chloride content (58.2 mg/l,
whereas the average composition of the other samples is around 5 mg/l). The latter is presumably from anthropic origin.

This figure indicates that both $CO_2$-rich and non $CO_2$-rich groundwaters have initially similar compositions, mainly driven by the local lithologies. The enrichment of $CO_2$ and bicarbonates in groundwater leads to an acidification of groundwater. Hence, $CO_2$-rich groundwaters are generally more aggressive and more mineralized. However, the waters of this dataset keep
the same relative proportions in cations composition, whatever their carbonate contents (Figure 4). This observation is not really in line with the hypothesis that carbonate dissolution explains the main origin of this dissolved $CO_2$. As discussed in Barros et al. 2021), this hypothesis was until now the most commonly accepted for the $CO_2$ origin. Indeed, if the presence of dissolved $CO_2$ in carbogazeous waters was due to carbonate dissolution, it would be expected that this groundwater would have also be enriched in $Ca^{2+}$ and possibly $Mg^{2+}$ ions, in comparison to non-carbogazeous waters, but this is not the case in
these Spa-Bru waters.

## 4.2 Constraining gas origin with helium and carbon isotopes

Inorganic carbon isotopes have proven to be a very powerful tool to make the distinction between different carbon sources. This isotopic proxy is particularly adapted to sparkling mineral waters (Fillimonova et al. 2020; Carreira et al. 2014; Redondo
and Yelamos 2004). However, the contribution of each carbon source is sometimes difficult to deconvolve. For example, a bulk C composition resulting from the mixing between marine limestones ($\delta^{13}C \approx 0$ ‰) and organic sediments ($\delta^{13}C \approx -20$‰) may have a $\delta^{13}C$ similar to that of mid-oceanic ridge basalts (MORB) ($\delta^{13}C \approx -6.5 \pm 2.5$ ‰), as shown by Sano and Marty



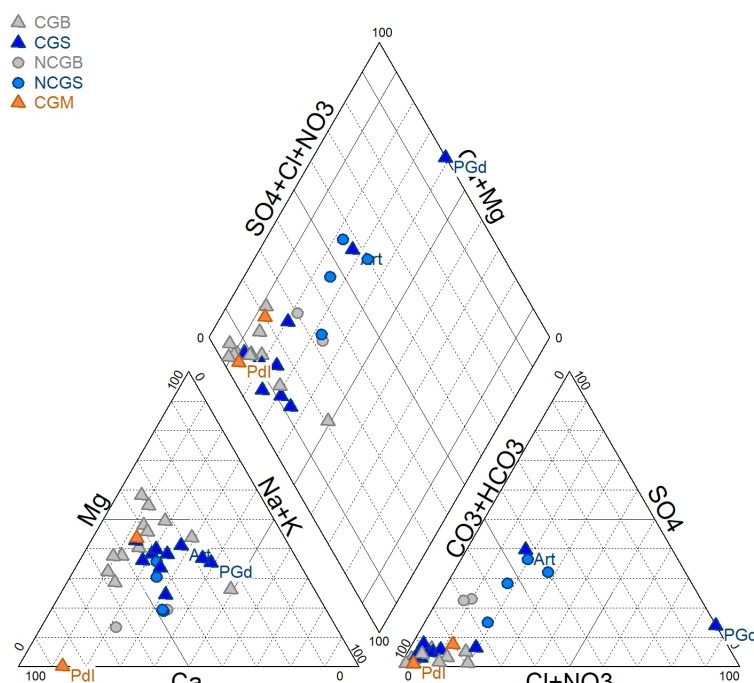

**Figure 4.** Piper diagram representing the relative proportion of cations and anions for each sample. CGB = carbogazeuos water from Bru area; CGS = carbogazeous water from Spa area, NCGB= non-carbogazeous water from Bru area, NCGS = non-carbogazeous water from Spa area, CGM = carbogazeous water from Mamedy area.





(1995) (Sano and Marty 1995). Atmospheric $\delta^{13}C$ is estimated around -8 ‰.

The $\delta^{13}C$ of inorganic carbon measured in our samples range between -8.8 ‰VPDB and -1.8 ‰VPDB for CG groundwater samples and between -27 ‰VPDB and -15.2 ‰VPDB for NCG groundwater samples. The distinction between CG and NCG groundwaters is clear: NCG groundwaters have much lighter carbon isotopic ratios. The majority of $\delta^{13}C$ values in $CO_2$-rich groundwater are clustered around the mantle MORB endmember, but some of them are also compatible with the range of the marine limestone endmember (Figure 5, where the different end-members are plotted according to Sano and Marty (1995)
(Sano and Marty 1995)).

The combination of $\delta^{13}C$ with the $CO_2$/$^3$He ratio permits to make this distinction, as shown in Figure 5: Indeed, a high $^3$He content is the signature of a mantle input as the production of $^3$He in the crust is negligible  (Andrews and Kay 1982). For a same concentration in dissolved $CO_2$, samples that are enriched in mantellic $CO_2$ have a lower $CO_2$/$^3$He. It can be observed
in Figure 5 that $CO_2$-rich groundwater samples have $CO_2$/$^3$He and $\delta^{13}C$ that are compatible with, (or very close to), the mantle MORB endmember. Their isotopic composition is very similar to the one observed in the Eifel area (*SQ, VQ* samples). Although the Spa-Bru CG groundwaters have $\delta^{13}C$ values quite close to a limestone source, their $CO_2$/$^3$He is 2 to 4 orders of magnitude lower than the limestone endmember. Their clear enrichment in $^3He$ demonstrates that the gas dissolved in the Spa-Bru groundwaters has a mantellic origin, probably from the nearby Eifel volcanic field.


### 4.3    Discriminating He origin with He and Ne isotopes

He isotopic ratio are normalized against the atmosphere isotopic composition and are expressed in $R/R_a$ (considering atmospheric $^3He/^4He = R_a = 1.382x10^{-6}$  (Sano and Fischer 2013)). In these waters, $R/R_a$ range between 0.92 and 2.70 ($\pm$
0.02). $^4He/^{20}Ne$ ratios are very variable with values ranging from 1.7 to 348.7 ($\pm$4%). Based on the measured $^4He/^{20}Ne$
ratios, and assuming all $^{20}Ne$ of atmospheric origin, the contribution of atmospheric He can be computed using mixing equation 1, where $^4He/^{20}Ne_{air}$ is considered equal to 0.267 according to  (Holocher et al. 2001) and $^4He/^{20}Ne_{mantle}$ is equal to 1000 according to  Dunai and Baur 1995.

$$\%He_{atm} = \frac{1 - \dfrac{^4He/^{20}Ne_{mantle}}{^4He/^{20}Ne_{sample}}}{1 - \dfrac{^4He/^{20}Ne_{mantle}}{^4He/^{20}Ne_{air}}} \tag{1}$$

This shows that atmospheric helium is negligible for all CG samples (less than 0.5 %) but more important for NCG samples (between 5 and 15 %). The measured $^3He/^4He$ ratios allow the computation of the origin of helium, which appears to be a mixture of crustal and mantle helium. The proportion of each source (crust and mantle) may be computed according to mixing





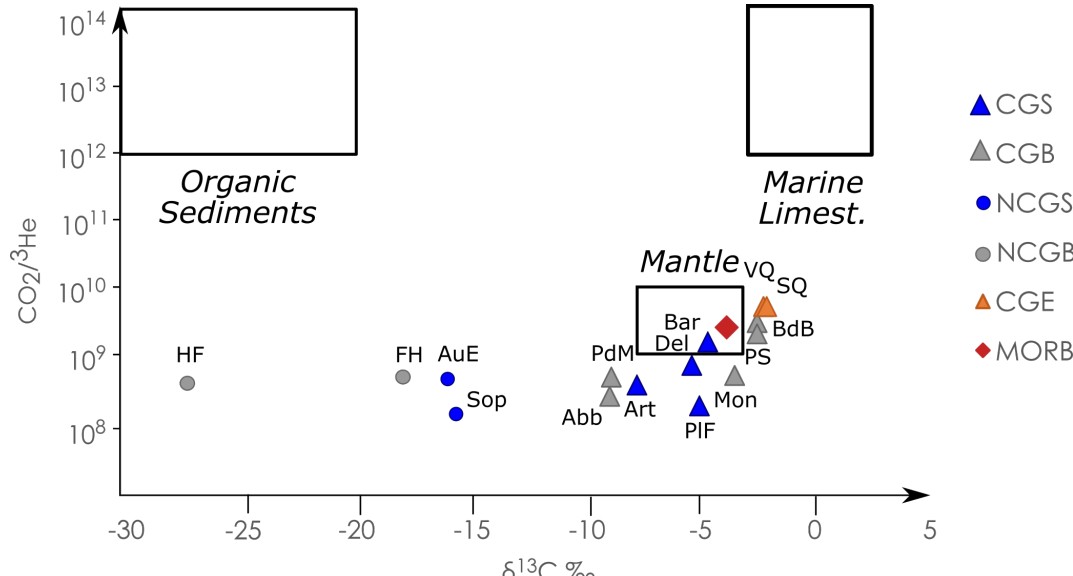

**Figure 5.** $CO_2$/$^3$He ratios vs $\delta^{13}C$ values for groundwater samples inrelation to mixing between the mantle, carbonate and organic $CO_2$ end-members based on Sano and Marty (1995) (Sano and Marty 1995). CGB = Carbogazeuos water from Bru area; CGS = carbogazeous water from Spa area, NCGB= non-carbogazeous water from Bru area, NCGS = non-carbogazeous water from Spa area, CGE = carbogazeous water from Eifel area. *VQ*, *SQ* and MORB are depicted from the values presented in Marty et al. (2020) (Marty et al. 2020). All $CO_2$-rich groundwater samples fall in or close to the mantle range. Non-$CO_2$-rich groundwater samples present much lighter C isotopic composition.

equations 2 and 3, where $^3He/^4He_{crust}$ and $^3He/^4He_{mantle}$ are taken equal to 0.02 and 6.5 respectively according to (Dunai and Baur 1995).


$$\%He_{mantle} = \frac{(1-\%He_{atm})(^3He/^4He_{crust} - ^3He/^4He_{sample})}{^3He/^4He_{crust} - ^3He/^4He_{mantle}} \qquad (2)$$

$$\%He_{crust} = 1 - \%He_{mantle} \qquad (3)$$

Crust-mantle mixing lines are also displayed in Figure 6.


Helium present in groundwater samples appears to be between 50 and 80 % from crustal origin. A distinction can be made between CG samples from Spa and Bru area, samples from Spa displaying more crustal helium (from 66 to 81%) than those of Bru (ranging from 54 to 57%). This may result from the local lithologies: Cambrian and Ordovician rocks from the Stavelot Massif are known for their high uranium content, often leading to high radon concentrations in cellars and underground





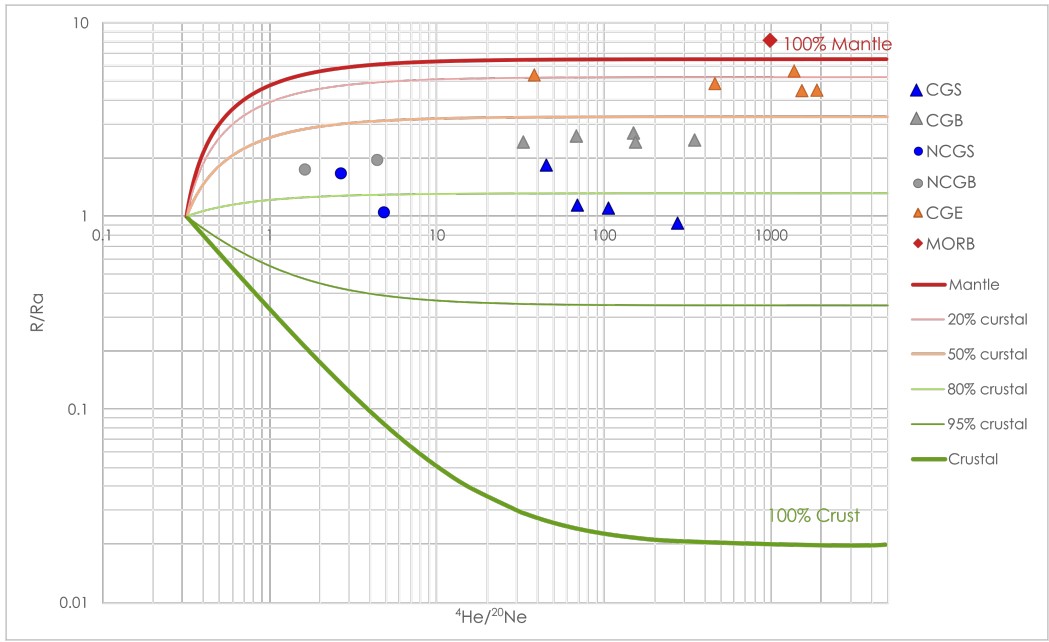

**Figure 6.** Proportion of crustal and mantle He in the samples, based on R/Ra and $^4He/^{20}Ne$ values. Mixing lines are computed from Equations 3 and 2. Values from Eifel and MORB are depicted from Marty et al. (2020)  (Marty et al. 2020). He atmospheric contribution is negligible. He crustal contribution appears to be more important for CG samples from the Spa area, as the local lithology is richer in uranium minerals.

buildings and also $^4He$ production through alpha-decay  (Depret et al. 2021; H.W. 2011). CG samples of Spa also differ from Eifel samples since they bear a larger proportion of crustal helium than the groundwaters of this volcanic region, where more than 80% of He is from the mantle.

# 5 Discussion

$CO_2/^3He$ and $\delta^{13}C$ measured in the samples from Eastern Belgium springs show that the dissolved $CO_2$ present in the springs from east Belgium originates from a mantle plume. The similar cation proportions for both CG and NCG groundwaters support the hypothesis that both groundwater types are initially the same (i.e.,having the same origin). This is at odds with the commonly accepted hypothesis according to which CG groundwater would travel at several kilometers in-depth to meet and dissolve carbonated layers, where it could have acquired its dissolved $CO_2$ content before upflowing rapidly to the surface. This hypothesis is also whittled by the fact that CG groundwaters are cold groundwaters, having similar temperatures to NCG groundwaters, between 12 and 13°C. Assuming a vertical temperature gradient of 30°C/km, this suggests that water circulation does not occur deeper than a few hundred of meters below the surface. $^3He/^4He$ ratios indicate that crustal $^4He$ enrichment occurred during the water circulation in the aquifer, but such addition of crustal fluid is impossible in less than 1 km flow water



paths.

The conceptual circulation model of water and gas at depth has then to be updated and is shown in Figure 7. The conceptual model was very similar to one of the models described by Pisolkar (2017) in his work aiming at developing integrated hydrogeological models representing the classical properties of $CO_2$-rich mineral waters systems in different contexts (Pisolkar n.d.).

Considering all our data with those of Barros et al. (2021) (notably the isotopic $\delta^{18}O$ and D/H composition of the water), 205 we propose here a revised model (Figure 7) to explain the origin of the gas-rich groundwaters of the Spa-Bru massif, involving the input of $CO_2$-$^3$He rich gas from the nearby Eifel massif through deep crustal faults.

The geomorphology of the system is mostly controlled by basin structures and anticlines. Slate beds act as low permeability barriers partitioning the aquifer and isolating the different compartments. Small faults and surface weathering enable the 210 infiltration and storage of water from the surface to the underground, whereas major deep-rooted faults act as $CO_2$ transport pathway from a degassing plume that can be located several tens of km away to the local aquifers. Where major faults reaching the surface do not encounter a sufficiently permeable and water-saturated zone (i.e. in the slates covered by a clayey colluvium as a result of weathering processes), $CO_2$-$^3$He rich dry gas are discharged to the surface and mofette are observed. The discharge points are mainly springs (at low topographic points or at geological low permeability/high permeability contacts) in 215 the hillslopes or water abstraction wells.

The $CO_2/^3$He and $\delta^{13}C$ composition of the samples is very close to samples taken in the Eifel Volcanic Fields (Figure 5, Table 1) and considering the proximity of the two sites (distance lower than 100 km), it seems very likely that the gas found in the Belgian springs comes from the degassing of the Eifel volcanic plume. This link between the two regions needs to be 220 further explained in terms of structural geology.

The main structural feature existing in the area are the Eupen thrust fault, the Xhoris thrust fault, and the normal faults linked to the opening of the Malmedy Graben (see Figure 1). All these faults could be connected at several kilometers in-depth to the Midi-Eifel thrust fault, a major thrust fault corresponding to the northernmost front of the Variscian Orogeny and connecting 225 both regions. This fault acts as an important seismic reflector and could thus be observed below the Stavelot Massif thanks to seismic measurements surveys lead by the DEKORP research group in the early nineties (Stiller et al. 1987).

# 6 Conclusions and future research

This paper answers a long-standing question regarding the origin of the dissolved $CO_2$ in the naturally sparkling mineral waters of eastern Belgium. The combination of $\delta^{13}C$ and $^3$He isotopes have shown with a high level of confidence that the dissolved 230 $CO_2$ in groundwater from the springs and in boreholes was from the mantellic origin, and can be very likely attributed to the

**Figure 7.** Updated conceptual model for the existence of CG and NCG groundwaters. Aquifer zones are located in the weathered parts of the bedrocks. The aquifer in fractured and slates beds isolate aquifers zones one from the other. Degassing $CO_2$ from a magmatic plume is brought to the system through regional thrust faults, and dissolved in groundwater.



degassing of the still buoyant neighboring Eifel volcanic plume, located at a distance lower than 100 km eastward. The role of the deep-rooted fractures that act as $CO_2$ transport pathway to the surface are still to be clarified, but several major thrust faults exist in the Rhenish Massif and could have connected the Eifel area with the studied area.

*Author contributions.* Agathe Defourny : Conceptualization, Methodology, Investigation, Writing – Original Draft, Visualization Pierre-Henri Blard : Validation, Resources, Funding Acquisition, Writing – Review  Editing Laurent Zimermann : Validation, Resources, Data Curation Patrick Jobé : Supervision Arnaud Collignon: Supervision, Resources Frédéric Nguyen : Supervision Alain Dassargues : Supervision, Project administration, Funding acquisition, Writing – Review  Editing

*Competing interests.* No competing interests are present in this manuscript.

*Acknowledgements.* This work is part of the ROSEAU project, as part of the Walloon program "Doctorat en Entreprise", co-funded by the SPW Région Wallonne of Belgium and the company Bru-Chevron S.A. (Spadel S.A.), under grant number 7984. The collaboration with the CRPG of Nancy was made possible thanks to the Europlanet Transnational Access Program, as this project has received funding from the European Union's Horizon 2020 research and innovation programm under grant agreement No 871149. The authors acknowledge these two institutions for their support.





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
