# Peer review of "$\delta^{13}C$ , $CO_2/^3$ He and $^3$ He/ $^4$ He ratios reveal the presence of mantle gas in the $CO_2$ -rich groundwaters of the Ardenne massif (Spa, Belgium)"

_Hydrology and Earth System Sciences, 2021_

## Author Response (AR1)

Dear Editor,

We are grateful to the 2 reviewers whose comments led us to revise our manuscript and hence clarify few imprecisions and issues that had been overlooked in the initial submission.

We notably revised the manuscript to clearly stand that:

- 1) our data do not permit to distinguish between a MORB and a plume endmember for the most pristine source of the helium component. We also added most recent references to state that the origin of the Eifel volcanism is more probably a MORB-like source (Moreira et al., 2018, Bekaert et al., 2019).
- 2) We also significantly revised section 5.2 in which we added a discussion about the different processed that may have fractionated the initial  $CO_2/{}^3He \delta^{13}C$  signature of the Spa and Bru gas rich waters. Note that this discussion does not modify our main parsimonious conclusion that points toward a volcanic (Eifel) origin for the Spa and Bru gaz rich sources.

You'll see below in our response letter how we addressed point-by-point all the other comments.

Editor decision: Publish subject to minor revisions (further review by editor) by Brian Berkowitz

**Comments to the author:**

The two reviewers appraised the manuscript very positively,but provided some important comments and insights that require careful consideration, particularly regarding the question of the volcanic origin of the gas. As you noted in your response, it is important to recognize that the origin of the volcanism of the Eifel is still the subject of debate, and that the work presented in the manuscript does not allow one to identify the actual origin. As also noted in your response, a brief discussion of isotopic fractionation of the different gases should be included. Given the nature of the review comments and your responses, I ask that you carefully revise the manuscript to thoroughly address the reviews. With a detailed Response Letter and careful revision of the manuscript, I expect that re-review will not be necessary, and that I will review the revised version.

**Reviewer 1**

Overall, the manuscript in quite well written and easy to follow. The research presented here brings further constraints on the source of gas in Belgium CO2-rich groundwaters. However, there is one important point that needs improvement in the paper. The authors keep referring to the Eifel mantle plume. However, the origin of the Eifel volcanic province is still matter of debate and the noble gas data from the Eifel area and more generally from the Central European Volcanic Province are not consistent with a plume origin but with an upper mantle source, i.e., similar signature as the MORB mantle (Moreira et al., 2018, Bekaert et al., 2019). The data in the present study cannot allow to distinguish between plume versus MORB source as the helium isotopic ratios are strongly influenced by crustal radiogenic production. But since the present data are consistent with previous CO2 and noble gas data from Eifel, it seems more consistent that the source of gas in the Belgium groundwater is the upper mantle. Therefore, I strongly recommend the authors to discuss this latter point in their paper, at least that the origin of the Eifel volcanic province is still debated, that a plume origin is not a consensus, and that their CO2 and noble gas data, while consistent with a mantle origin (which is the key aspect of this study), cannot be used to distinguish plume vs MORB. In particular, I would suggest to change the wording in the abstract ("buoyant Eifel mantle plume"), and all the reference to the Eifel "plume" (as in Figure 2, lines 78-82, 190, 219, 231, etc).

Indeed, the geodynamic origin of the Eifel volcanic field is not definitely understood and we agree that our dataset does not permit to distinguish between a MORB or a plume origin. We thus revised the text to only mention a "mantle" or "volcanic" source, without distinction between a MORB or a plume source.

Minor comments:

Lines 22-29, 57-64: References are missing.

Line 65: typo, volcanic

Line 98-99: problem format of references, this is the case in several other places (e.g., line 144)

Line 117: 10 per cent Line 119: samples located, please correct Captions Figures 4, 5. Carbogazeous; in relation

All these typos have been corrected.

Line 130-135: This should be moved to the discussion, this is not part of the results.

OK, done.

**Equation 1: In fact, He in the samples is a mix of 3 components (air, crust and mantle) so this equation is not really valid.**

Given that the crustal and mantle endmembers have  ${}^{4}\text{He}/{}^{20}\text{Ne}$  ratios that are much higher than the one of atmosphere and are near similar, we assumed here that the non atmospheric source of  ${}^{4}\text{He}$  is a single mantle-crust indistinguishable endmember. This assumption makes the equation (1) to be valid.

**Line 173: The value of 6.5 is for the subcontinental lithospheric mantle, not for the upper mantle. Please precise this for consistency, as you keep talking either about plume or MORB.**

Now read here: "We chose this 6.5 Ra "mantle" endmember as representative of the subcontinental lithospheric mantle (Dunai and Baur, 1995)."

**Reviewer 2**

This paper by Defourny et al. reported new data of hydrochemical,  $\delta^{13}$ C, and  ${}^{3}$ He/ ${}^{4}$ He for water and gas samples from the Ardenne Massif (Belgium) and presented a model to establish a link between the CO2-rich groundwater and the magma source beneath the Eifel volcanic field in Germany. As pointed out by Referee #1, the authors should be careful to propose a mantle plume origin for the gases. In addition, the influence of He-CO2 fractionation and carbon isotope fractionation should be evaluated in the discussion part. Overall, the manuscript is well written, and I support the publication of this work after minor revision based on the comments below.

General points.

1. In the title, the authors seem to only emphasize CO2/3He ratios as evidence for the presence of mantle gases (e.g., CO2 and He). However, in the main text, the authors also mentioned  $\delta$ 13C and 3He/4He evidence; especially 3He/4He is more direct in tracing the release of mantle He and CO2 (commonly interpreted as the carrier gas for He). Therefore, I suggest an appropriate revision of the title based on the integrated lines of evidence presented in this study.

We revised the title into: " $\delta^{13}C$ ,  $CO_2/^{3}He$  and  $^{3}He/^{4}He$  ratios reveal the presence of mantle gas in the CO2-rich groundwaters of the Ardenne massif (Spa, Belgium)"

2. It may be better to show some data (e.g., the  $\delta$ 13C and 3He/4He values) in the abstract to make it easier for the readers to get some detailed information.

Agreed, we now quote the range of these data in the abstract.

**3.** Overall, the results section should be appropriately shortened by simply reporting the data and not going too much into data interpretation. Some sentences or paragraphs (e.g., Lines 130-135 and 138-143) in the Results can be moved to the discussion part. In addition, the titles of sub-sections 4.1, 4.2, and 4.3 are commonly used in the discussion of a paper.

We followed this recommendation: we shortened the Results section and moved several sections in the discussions.

4. In the discussion part, it should be noted that  $CO_2$  and He have different solubilities in melts and water, and  $CO_2/{}^3$ He ratios are easy to be fractionated from their original values due to magma degassing, hydrothermal degassing, or calcite precipitation (see details in Ray et al., 2009, Chemical Geology). Additionally, the influence of carbon isotope fractionation should also be evaluated for the  $\delta 13C$  data.

We are grateful to the reviewer for having raised this point. We added few sentences in section to discuss these potential processes that may have modified the  $\delta^{13}$ C and the CO2/3He ratios of the initial source gas. We also added on Figure 5 the fractionation lines that shows the effects of calcite precipitation at three different temperatures (50, 100 and 150°C), if this precipitation occurred in an open system and hence led to a Rayleigh distillation.

"Although most of the CG waters are close to the mantle endmember in this  $CO_2^{\beta}He$  vs  $\delta^{3}C$  space, some data points do not perfectly match the mixing lines between mantle and organic or carbonate endmembers. Hence, other processes may have fractionated the initial  $CO_{\gamma}^{\beta}He$  vs  $\delta^{3}C$  need to be considered (e.g. Ray et al., 2009; Barry et al., 2020). CO2 and helium having different solubilities, partial degassing may fractionate the  $CO_{2/3}He$  ratio. However, the solubility of  $CO_{2}$  being larger than the one of helium, this process should lead to increase the  $CO_2^{\beta}He$  of waters affected by degassing ratio, contrary to what we observe here. Moreover, there is almost no correlation between the  $CO_2^{\beta}He$ ratios and the dissolved helium concentrations (after correction for atmospheric helium), an observation that makes this process unlikely (Table 1). Another physical process that has the ability to modify the initial  $CO_2/^3$ He- $\delta^{13}C$  signature is the precipitation of calcite, leading to lower the  $\delta^{13}C$  and the  $CO_2/^3$ He by a Rayleigh distillation (in open system). In figure 6, following (Ray et al., 2009; Barry et al., 2020), we modeled the effect of calcite precipitation at various temperatures (50°C, 100°C and 150°C), assuming an initial gas composition similar to the Eifel endmember (Braüer et al., 2013). Although this process may, is theory, explain part of the scatter observed within the carbogazeous sources of Spa and Bru, we should however be cautious because calcite precipitation is not evident in the underlying rocks where these fluids are supposed to have transited. It is nevertheless important to

stress that this fractionation remains limited since it does not hamper the identification of a clear mantle signature.

The low 13C and  $CO_2 \beta^3$ He values of the noncarbogazeous sources stand below the pure mixing curve between the mantle and organic carbon endmembers (Fig. 6). However, our geochemical dataset (see section 5.1) shows that these gas poor fluids are not derived from the same water than the carbogazeous sources. Hence, this  $\delta^{13}C$  and  $CO_2$  depletion of noncarbogazeous probably result from another initial endmember, that could result from a previous mixing between two fluids with mantle and organic carbon compositions (Fig. 5).

Other points.

Line 41. The reference citation style should be corrected as that will appear in the final printed version. Similar problems are also found in other places of the main text (e.g., Lines 74-75, 80, 109, and 142-143).

Line 87. Better to replace traces with trace, which is more frequently used in literature. Same problem in Line 91.

Line 118. Typo. It should be CGS samples.

Line 122. PIG or PGd? Table 1 and Figure 4 show that the sample with Cl content of 58.2 mg/L is PGd.

Line 143. A reference is needed for the  $\delta$ 13C value of the atmosphere.

Line 163. Usually, the corrected 3He/4He ratios should also be reported for the samples considering contamination by air-derived helium. The correction method was proposed by Craig et al. (1978) and summarized in detail by Hilton (1996, Chemical Geology).

Line 166. A 4He/20Ne ratio of 0.318 is recommended for the atmosphere, according to Sano and Wakita (1985, JGR).

Lines 170-179. The proportions of air, mantle and crust can be calculated following Sano

All these typos have been corrected.